# CausalCite: A Causal Formulation of Paper Citations

## Abstract

Evaluating the significance of a paper is pivotal yet challenging for the scientific community. While the citation count is the most commonly used proxy for this purpose, they are widely criticized for failing to accurately reflect a paper's true impact. In this work, we propose a causal inference method, SynMatch, which adapts the traditional matching framework to high-dimensional text embeddings. Specifically, we encode each paper using the text embeddings by large language models, extract similar samples by cosine similarity, and synthesize a counterfactual sample by the weighted average of similar papers according to their similarity values. We apply the resulting metric, called CausalCite, as a causal formulation of paper citations. We show its effectiveness on various criteria, such as high correlation with paper impact as reported by scientific experts on a previous dataset of 1K papers, (test-of-time) awards for past papers, and its stability across various sub-fields of AI. We also provide a set of findings that can serve as suggested ways for future researchers to use our metric for a better understanding of a paper's quality.[1]

## 1 Introduction

Recent years have seen explosive growth in the number of scientific publications, making it increasingly challenging for scientists to navigate the vast landscape of scientific literature. Therefore, identifying a good paper has become a crucial challenge for the scientific community, not only for technical research purposes, but also for decision-making, such as funding allocation (Carlsson, 2009), research evaluation (Moed, 2006), recruitment criteria (Gary Holden & Barker, 2005), and university ranking and evaluation (Piro & Sivertsen, 2016).

A traditional method to recognize paper quality is peer review, a mechanism that leads to a large number of accepted papers, underscoring the challenge of identifying truly impactful research. The randomness and flaws inherent in the peer review process further exacerbate this challenge (Cortes & Lawrence, 2021; Rogers et al., 2023; Shah, 2022; Prechelt et al., 2018; Resnik et al., 2008). Another commonly used metric is citations. However, this metric faces criticism for biases, such as a preference for survey, toolkit, and dataset papers (Zhu et al., 2015; Valenzuela-Escarcega et al., 2015). Citations, together with altmetrics (Wilsdon et al., 2015), which incorporates the social media attention to a paper, often have biases towards papers with extensive publicity and promotion or those authored by established figures in the field.

To provide a more equitable assessment of paper quality, we employ the causal inference framework (Hernán & Robins, 2010) to quantify a paper's impact by how much of the academic success in the follow-up papers should be *causally attributed* to this paper. We introduce CausalCite, an enhanced version of citations that poses the following *counterfactual question*: "*had this paper never been published, what would be the impact on this paper's current follow-up studies?*" To compute the causal attribution for each follow-up paper, we synthetically construct a counterfactual paper that addresses a similar topic but is not built on the paper of interest. Our approach allows CausalCite to quantify scientific impact through a causal lens, offering an alternative understanding of a paper's impact within the academic community.

To tackle this problem, we need to address its main challenge – performing causal inference on high-dimensional text data. Leveraging the current advancement of large language models (LLMs),

---

[1]Our code and data are uploaded to the submission system and will be open-sourced upon acceptance.

we adapt causal inference methods to the high-dimensional space created by LLMs, marrying the causal inference and natural language processing (NLP) domains. Specifically, we propose SYN-MATCH, a new method which uses LLMs to encode an academic paper as a high-dimensional text embedding and then operationalizes the matching method by a combination of techniques in information retrieval (Manning et al., 2008) and semantic text similarity matching (Majumder et al., 2016; Chandrasekaran & Mago, 2022), such as BM25 (Robertson & Zaragoza, 2009) and cosine similarity matching (Majumder et al., 2016). After finding close matches for each paper, we leverage all the close matches to synthesize a counterfactual sample by a weighted average according to their similarity scores. This enables a stable causal effect estimation. Finally, we quantify the counterfactual causal effect as the difference between the synthesized control sample and the original sample.

We conduct extensive experiments utilizing the Semantic Scholar corpus (Lo et al., 2020), which comprises 206M papers and 2.4B citation links. We empirically validate CAUSALCITE by showing higher predictive accuracy of paper impact (as judged by scientific experts on a past dataset of 1K papers (Zhu et al., 2015)) compared to citations and other previous impact assessment metrics. We further show a stronger correlation of the metric with the Test-of-Time (ToT) paper awards. Furthermore, we show that, unlike citation counts, our metric exhibits a greater balance across various research domains in AI, e.g., General AI, NLP and Computer Vision (CV). While citation numbers for papers in those domains vary significantly – for example, while an average CV paper has many more citations on average than a NLP paper, our Causal Impact Index scores papers across AI subfields more similarly. After demonstrating the desirable property of our metric, we present several case studies of its applications. Our evaluations reveal that the quality of conference best papers is noisier on average than that of ToT papers (Section 7.1). We find that our CAUSALCITE follows a power law distribution (Section 7.2), and conduct a case study showcasing CAUSALCITE for some renowned papers (Section 7.3). Finally, we utilize our metric to identify high-quality papers that are less recognized by citation counts (Section 7.4).

**Contributions.** (1) We introduce CAUSALCITE, a counterfactual causal effect-based formulation for paper citations. (2) We introduce SYNMATCH, a novel method that leverages LLMs and causal inference to estimate the counterfactual causal effect of a paper. (3) We conduct comprehensive analyses, encompassing three performance evaluations and four findings using our new metric.

## 2 PROBLEM FORMULATION

### 2.1 GRAPH NOTATIONS

Our problem formulation involves a citation graph and a causal graph. We use lowercase letters for specific papers and uppercase for an arbitrary paper treated as a random variable.

**Citation Graph** In the citation graph $\mathbb{G} := (\mathbb{P}, \mathbb{L})$, $\mathbb{P}$ is a set of papers, and each edge $\ell_{i,j} \in \mathbb{L}$ indicates that an earlier paper $p_i$ influences (i.e., is cited by) a follow-up paper $p_j$. To obtain the citation graph, we use the Semantic Scholar Academic Graph dataset (Kinney et al., 2023) with 206M papers and 2.4B citation edges.

**Causal Graph** The causal graph, shown in Figure 1, highlights the contribution of a paper $a$ to a follow-up paper $b$. It uses a binary variable $T$ to indicate if $a$ influences $b$ and an effect variable $Y$ to represent the success of $b$. We use $\log_{10}$ of citation to quantify $Y$, although other scoring metrics are applicable. In the causal graph, the source of randomness originates from paper $b$, and the random variables pertain to properties of paper $b$. We introduce two sets of variables in this causal graph: (i) The set of confounders, which are the common causes of $T$ and $Y$. For instance, the research area of $b$ impacts both the likelihood of a paper citing $a$ and its own citation count. (ii) Descendants of the treatment, comprising mediators (e.g., paper $a$ influencing the quality of paper $b$ and subsequently influencing its citations) and colliders (e.g., both the influence from $a$ and the citations of $b$ itself influencing later awards received by $b$).

### 2.2 CAUSALCITE INDICES

In this section, we introduce various indices that measure the causal impact of a paper.

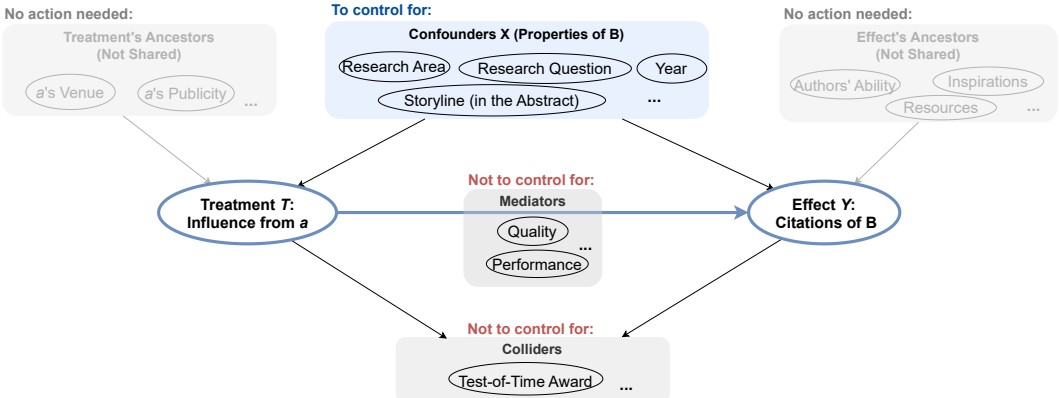

Figure 1: the causal graph of our study. We are interested in the treatment effect of the influence from $a$ to the resulting citations of $B$ (outlined in blue). To obtain this causal effect, we consider three sets of variables: the **confounder set** which we need to control for (colored in the blue area), the **mediator and collider set** which we need to explicitly exclude (colored in gray with the red text "Not to control for"), and the **non-shared ancestor set** which does not need any action (colored in gray with the gray text "No action needed").

**Two-Paper Interaction: Pairwise Causal Impact (PCI).** To examine the causal impact of a paper $a$ on a follow-up paper $b$, we define the pairwise causal impact $\text{PCI}(a, b)$ by unit-level causal effect:

$$\text{PCI}(a, b) \coloneqq y^{t=1} - y^{t=0} , \tag{1}$$

where we compare the outcomes $Y$ of the paper $b$ had it been influenced by paper $a$ or not, denoted as the actual $y^{t=1}$ and the counterfactual $y^{t=0}$, respectively. Note that the counterfactual $y^{t=0}$ can never be observed, but only estimated by statistical methods, as we will discuss in Sections 3 and 4.

**Single-Paper Quality Metrics: Total Causal Impact (TCI) and Average Causal Impact (ACI).** Let $\boldsymbol{S}$ denote the set of all follow-up studies of paper $a$. We define total causal impact $\text{TCI}(a)$ as the sum of the pairwise causal impact index $\text{PCI}(a, b)$ across all possible $b$'s. That is,

$$\text{TCI}(a) \coloneqq \sum_{b \in \boldsymbol{S}} \text{PCI}(a, b) . \tag{2}$$

This definition provides an aggregated measure of a paper's influence across all its follow-up papers.

As the causal inference literature is usually interested in the average treatment effect, we further define the average causal impact (ACI) index as the average per paper PCI:

$$\text{ACI}(a) \coloneqq \frac{\text{TCI}(a)}{|\boldsymbol{S}|} = \frac{1}{|\boldsymbol{S}|} \sum_{b \in \boldsymbol{S}} \left( y^{t=1} - y^{t=0} \right) . \tag{3}$$

To establish its causal linkage more explicitly, we note that $\text{ACI}(a)$ is equal to the average treatment effect on the treated (ATT) of paper $a$ (Rosenbaum & Rubin, 1983).

## 3 CLASSICAL CAUSAL INFERENCE METHODS

In this section, we provide a brief overview on existing methods in causal effect estimation and describe their limitations on our research question.

**Blocking the Backdoor Paths.** It is important to control for the right set of variables for causal effect estimation. For example, in CAUSALCITE, we control the domain of the paper to estimate causal impact. Specifically, (Pearl, 1995) shows one need to block all backdoor paths from the treatment to the effect variable, allowing only the association along the $T \rightarrow Y$ path. Hence, we want to control for certain factors, such as the research topic across the treatment and control groups.

Using principles in do-calculus (Pearl, 1995), such variables to control for must *include all the confounders*, as marked in the blue box in Figure 1, but *exclude the descendant variables* of the treatment, as marked in the gray boxes with the red note "Not to control for" in Figure 1. As a note, if we unintentionally control for descendants of the treatment, such as the paper's reported model

performance, then we might get an undesired control group, such as papers that do not build on $a$ but still achieve as good performance as those that build on $a$ (Imagine a paper that did not build LLMs but still achieves the same performance as LLM-based models, then this paper must be extra strong, deviating from the normal paper in the control group). Technically, by avoiding the descendants of the treatment, we prevent undesired effects, such as blocking mediators or accounting for colliders, which will result in erroneous inference.

**Randomized Control Trials (RCTs).** The ideal way to obtain causal effects is through randomized control trials (RCTs). For example, when testing a drug, we randomly split all patients into two groups, the control group and the treatment group, where the random splitting ensures the same distribution of the confounders across the two groups such as gender and age. However, RCTs are usually not easily achievable, in some cases too expensive (e.g., tracking hundreds of people's daily lives for 50 years), and in other cases unethical (e.g., forcing a random person to smoke), or infeasible (e.g., getting a time machine to change a past event in history).

For our research question on a paper's impact, utilizing RCTs is impractical. Because it is unfeasible to randomly divide researchers into two groups, instructing one group to base their research on a specific paper $a$ while the other group does not, and then observe the citation count of their papers years later.

**Ratio Matching** In the absence of RCTs, matching serves as an alternate method for determining causal effects from observational data. In this case, we can let the treatment assignment happen naturally, such as taking the naturally existing set of papers and running causal inference by adjusting for the variables that block all paths. Given a set of naturally observed papers, one of the most commonly used causal inference methods is ratio matching (Rosenbaum & Rubin, 1983), whose basic idea is to iterate over all possible values $x$ of the adjustment variables $X$ and obtain the difference between the treatment group $\mathcal{T}$ and control group $\mathcal{C}$.

While ratio matching is practical when there is a small set of values for the adjustment variables to sum over, its applicability dwindles with high-dimensional variables like text embeddings in our context. This scenario may generate numerous intervals to sum over, presenting numerical challenges and potential breaches of the positivity assumption.

**One-to-One Matching** To handle the adjustment variables in the high-dimensional space, one possible way is to avoid pre-defining all their possible intervals. but instead, iterate over each unit in the treatment group to match for its closest control unit. Consider a given follow-up paper $b$, and a set of candidate control papers P, where each paper $p_i$ has a citation count $y_i$, and vector representation $t_i$ of the confounders (e.g., topic, storyline, etc). The one-to-one matching method estimates PCI as

$$\widehat{\text{PCI}}(a,b) = y_b - y_{\text{argmax}_{p_i \in P} m_i} = y_b - y_{\text{argmax}_{p_i \in P} \text{sim}(t_y, t_i)} , \quad (4)$$

where we approximate the counterfactual sample by the paper $p_i \in P$ which is the most similar to paper $b$ by the matchedness score $m_i$, which is obtained by the cosine similarity $\text{sim}$ of the confounder vectors. A limitation of this one-to-one matching method is that it might induce large instability in the result, as only taking one paper with similar contents may have a large variance in citations when the matched paper slightly differs.

## 4  SYNMATCH: STABILIZING MATCHING BY SYNTHESIS

### 4.1  THE SYNMATCH METHOD

To fill in the gap of the existing matching methods, we propose SYNMATCH, which mitigates the instability issue of the one-to-one matching by replacing it with a convex combination of a set of matched samples to form a synthetic counterfactual sample. Specifically, we identify a set of papers $p_i \in P$ with high matchedness scores $m_i$ to the paper $b$, and synthesize the counterfactual sample by an interpolation of them:

$$\widehat{\text{PCI}}(a,b) = y_b - \sum_{p_i \in P} w_i y_i = y_b - \sum_{p_i \in P} \frac{m_i}{\sum_i m_i} y_i , \quad (5)$$

where the weight $w_i$ of each paper $p_i$ is proportional to the matchedness score $m_i$ and normalized.

The contributions of our method are as follows: (1) we adapt the traditional matching methods from low-dimensional covariates to any high-dimensional variables such as text embeddings; (2) different

from the ratio matching, we do not stratify the covariates, but synthesize a counterfactual sample for each observed treated units; (3) due to this iteration over each treated unit instead of taking the population-level statistics, we closely control for exogenous variables for the ATT estimation, which circumvents that need for the structural causal models; (4) we further stabilize the estimand by a convex combination of a set of similar papers. Note that the contribution of Eq. (5) might seem to bear similarity with synthetic control, but they are fundamentally different, in that synthetic control runs on time series, and fit for the weights $w_i$ by linear regression between the time series of the treated unit and a set of time series from the control units, using each time step's values in the regression loss function. Therefore, our SYNMATCH proposal is novel and constitutes a crucial step to bridge the traditional matching methods to high-dimensional text embeddings.

## 4.2 OVERALL ALGORITHM

Using SYNMATCH as the backbone theoretical method, we develop the overall algorithm to calculate the causal impact indices as follows.

**Obtaining the Control Variables** Given the two papers $a$ and $b$, we first collect the set of confounders using the causal graph in Figure 1: paper $b$'s publication year, research area, research question, and storyline. When collecting the confounders, we have direct access to the year variable, but for the remaining ones, we unleash the power of NLP to represent them by text embeddings in a rich, high-dimensional space, instead of manually coding them as categorical variables. We describe the process below. Starting from any given paper $b$ with its title and abstract, we not only need to model these contents, but also need to pay attention not to accidentally control for any descendant of the treatment, such as the quality/performance (through expressions such as "we achieved 90% accuracy"). Therefore, we first exclude expressions about the quality or performance of the paper, such as the notion "state-of-the-art" and the exact performance numbers in arabic numbers. Then, we concatenate the title and abstract to encode them into a text variable $t_b$ later for the matching.

**Selecting the Text Encoder** When projecting the text into the vector space, we need a text encoder with a strong representation power for scientific publications, and is sensitive towards two-paper similarity comparisons regarding their research topics, questions, and storylines. For the representation power for scientific publications, instead of general-domain models such as BERT (Devlin et al., 2019) and RoBERTa (Liu et al., 2019), we consider LLM variants pretrained on large-scale scientific text, such as SciBERT (Beltagy et al., 2019b), SPECTER (Cohan et al., 2020), and MPNet (Song et al., 2020).

To check the capability for two-paper similarity comparisons, We conduct a small-scale empirical study comparing human-ranked paper similarity and model-identified semantic similarity. We find that MPNet correlates the best with human judgments, and give more distinct scores to papers with different levels of similarity. This capability advantage may be attributed to its Siamese network objectives in the training process (Song et al., 2020). See more details in Appendix A.2.

**Enabling the Matching Method at Scale** Since we use one of the largest available paper database, the Semantic Scholar dataset (Kinney et al., 2023) of 206M papers with 2.4B citation links, we need to optimize our algorithm for large-scale paper matching. The vast paper size is a critical problem, as for example, even after we filter by the publication year, the number of papers in the same year could be up to 8.8M, and on average 7.5M per year since 2010.

In order to conduct text matching across millions of papers, we use the combination of two NLP tasks, first information retrieval (IR) (Manning et al., 2008) and semantic textual similarity (STS) (Majumder et al., 2016; Chandrasekaran & Mago, 2022). IR is used for large-scale, efficient document-level text retrieval, an example of which is Google search, and we use the most commonly used IR method, BM25 (Robertson & Zaragoza, 2009), as the preparation step for the text matching. Briefly, it is a bag-of-words retrieval function that uses term frequencies and document lengths to estimate relevancy between two text documents. Deploying this method, we can find a set of candidate papers for, for example, two million papers, at a speed 250x faster than the text embedding cosine similarity matching.

We develop a first-coarse-then-fine-grained approach, which first obtains a broad set of 100 candidate papers using BM25, and then runs the fine-grained matching by cosine similarity. In the cosine similarity matching process, we use the MPNet model to encode the text of each paper $p_i$ into an

embedding $t_i$, with which we get the matchedness score $m_i$ according to Eq. (4), and the normalized weight $w_i$ by Eq. (5).

**Deriving the Causal Impact Indices by Our Novel Numerical Estimation Method** After using the above method to obtain each $\mathrm{PCI}(a, b)$, we aggregate them to obtain the ACI and TCI in Algorithm 1.

---

**Algorithm 1** Algorithm to derive the causal impact index

---

1: **Input**: Paper $a$, all its follow-up papers $\boldsymbol{B}$, a random subset $\boldsymbol{B}'$, and non-follow-up papers $\boldsymbol{C}$.
2: **procedure** GETACI($a$)
3:     ACI $\leftarrow 0$
4:     **for** each $b_i$ in $\boldsymbol{B}'$ **do**
5:         $I_i \leftarrow \mathrm{getPCI}(a, b_i, \boldsymbol{C})$
6:         ACI $\leftarrow$ ACI $+ \frac{1}{|\boldsymbol{B}'|} \cdot I_i$
7:     **end for**
8:     TCI $\leftarrow$ ACI $\cdot |\boldsymbol{B}|$
9:     **return** ACI and TCI
10: **end procedure**

---

Given the vast paper numbers mentioned above, there is also a critical challenge when aggregating the total causal impact because the number of follow-up papers for a study can be up to tens of thousands, such as the 57,200 citations for the ImageNet paper (Deng et al., 2009). To address this problem, we innovate a numerical estimation method using a carefully designed random subset of the follow-up papers.

A naive method to achieve this is through Monte Carlo (MC) sampling. However, the basis assumption behind MC sampling is the law of large numbers (LLN), but unfortunately, it requires a very large sample size when it comes to long-tailed distributions, which is the usual case of citations. Since citations are very likely to be concentrated in the head part of the distribution, we will not be able to afford the computers of budget for a huge sample size that covers the narrow head area. Instead, we propose a novel numerical estimation method for the follow-up paper sampling, inspired by importance sampling (Singh, 2014; Kloek & van Dijk, 1976).

Our numerical estimation method works as follows. First, we propose the formulation that the relation between ACI and the TCI is an integral over all possible paper $b$'s. Then, we formulated the above sampling problem as integral estimation or area-under-the-curve estimation. We draw inspiration from Simpson's method, which estimates integrals by binning the input variable into small intervals. Analogously, we bin the large set of follow-up papers into $n$ equally-sized intervals and perform random sampling over each bin, which we then sum over. In this way, we make sure that our samples come from all parts of the long-tailed distribution and are a more accurate numerical estimate for the actual TCI.

## 5 EXPERIMENTAL SETUP

**Dataset** We use the Semantic Scholar dataset (Lo et al., 2020; Kinney et al., 2023)[2] which includes a paper corpus of 206M scientific papers, and a citation graph of 2.4B+ citation edges. For each paper, we obtain the title and abstract for the matching process. We list some more details of the dataset in Appendix B, such as the number of papers reaching 8M per year after 2012.

**Implementation Details** For the text embedding model, we compared SciBERT (Beltagy et al., 2019a), SPECTER (Cohan et al., 2020), and MPNet (Song et al., 2020), among which we find that MPNet performs the best, outperforming the next one by 1.12 points on our annotated dataset described in Appendix A.2. We deploy the *all-mpnet-base-v2* checkpoint of the MPNet using the *transformers* Python package (Wolf et al., 2020), and set the batch size to be 32. For the set of matched papers, we consider papers with cosine similarity scores higher than 0.93, and take the top ten papers above the threshold. To enable efficient operations on the large-scale citation graph, we use the Dask framework[3] which optimizes for data processing and distributed computing. We optimize

---

[2]https://api.semanticscholar.org/api-docs/datasets
[3]https://www.dask.org/

our program to take around 100GB RAM, and on average 20-30 minutes for each $\text{PCI}(a, b)$ after matching against up to millions of candidates. More implementation details are in Appendix A.3.

## 6 PERFORMANCE EVALUATION

The contribution of a paper is inherently multi-dimensional, making it infeasible to encapsulate its richness fully through a scalar. Yet the demand for a single, comprehensible metric for research impact persists, fueling the continued use of traditional citations despite their known limitations. In this section, we show how our new metrics significantly improve upon traditional citations by providing quantitative evaluations comparing the effectiveness of citations, Semantic Scholar's highly influential (SSHI) citations (Valenzuela-Escarcega et al., 2015), and our CAUSALCITE metric.

### 6.1 AUTHOR-IDENTIFIED PAPER IMPACT

In this experiment, we follow the evaluation setup in Valenzuela-Escarcega et al. (2015) to use an annotated dataset (Zhu et al., 2015) comprised of 1037 papers, annotated according to whether they serve as significant prior work for a given follow-up study. Although paper quality evaluation can be dauntingly tricky, this dataset was cleverly annotated by first collecting a set of follow-up studies and letting one of the authors of each paper go through the references they cite and select the ones that significantly impact their work. This approach ensures that on average, for each paper $b$, there exists 33 annotated papers, each annotated as whether $b$ significantly impacts them or not.

For each annotated paper, we assess how well citations, SSHI citations, and CAUSALCITE are aligned with human annotations. For each of the metrics, a paper is aligned with human annotations when a significant paper's value by this metric is higher than that of a non-significant one, and a violation is vice versa. Accordingly, we can compute the accuracy of each of the metrics, which are reported in Table 1. This table shows that our metric has the highest accuracy.

| Metric | Accuracy |
|---|---|
| Citations | 71.33 |
| SSHI Citations | 75.25 |
| CAUSALCITE | **80.29** |

Table 1: Accuracy of existing citation metrics and our CAUSAL-CITE over 1K papers annotated by Zhu et al. (2015).

### 6.2 TEST-OF-TIME PAPER ANALYSIS

The Test of Time (ToT) paper award is a prestigious honor bestowed upon papers that have made substantial and enduring impacts in their field. In this section, we use a dataset of over 400 papers, including 72 ToT papers. This dataset is sourced from ten leading AI conferences spanning general AI (NeurIPS, ICLR, ICML, and AAAI), NLP (ACL, EMNLP, and NAACL), and CV (CVPR, ECCV, and ICCV).[4] To contrast the ToT papers, we consider a control group of 10 randomly selected non-ToT papers for each conference and year. Our corpus includes 44 conference-year paper on average.

In Table 2, we show the correlations of various metrics with the ToT awards. In this table, CAUSALCITE achieves the highest correlation of 0.623, which is $+31.99\%$ better than that of citations. Furthermore, we visualize the correspondence of our metric and ToT in Figure 2. We can see that the CAUSALCITE distributions of ToT papers and non-ToT papers show a substantial difference in Figure 2a. We also show three examples of ToT papers in Figure 2b, where the ToT papers differ from the non-ToT papers by one or two orders of magnitude.

| Metric | Corr. Coef. |
|---|---|
| Citations | 0.472 |
| SSHI Citations | 0.317 |
| TCI | **0.623** |

Table 2: Correlation coefficients of each metric and ToT paper award. We use the Point Biserial Correlation (Tate, 1954), a special case of Pearson correlation between a continuous variable and a dichotomous variable.

### 6.3 TOPIC INVARIANCE OF CAUSALCITE

A well-known issue with citations is their inconsistency across different fields. What might be considered a large number of citations in one field might be seen as average in another. In contrast, we show that our ACI index

| Research Area | ACI | Citations | SSHI Citations |
|---|---|---|---|
| General AI (n=16) | 0.748 | 2,024 | 267 |
| CV (n=36) | 0.734 | 7,238 | 1,088 |
| NLP (n=20) | 0.763 | 1,785 | 461 |

Table 3: The average of each metric by research area.

does not suffer from this issue. We show this using our ToT dataset, where we control for the quality of the papers to be ToT but vary the domain by the three fields: general AI, CV, and NLP. We

---

[4]We get this list by selecting the top conferences on Google Scholar using the h5-Index ranking in each of the above domains: general AI (link), CV (link), and NLP (link).

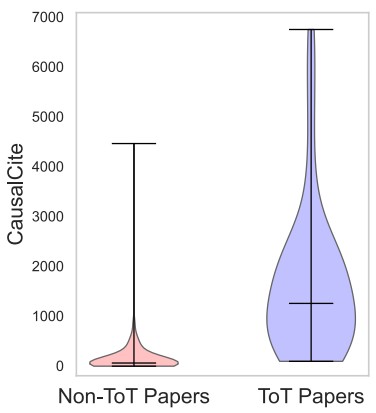
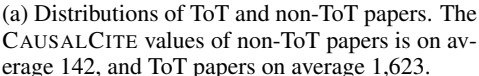

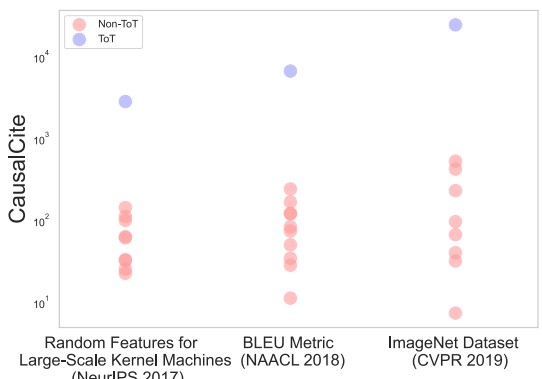

(a) Distributions of ToT and non-ToT papers. The CAUSALCITE values of non-ToT papers is on average 142, and ToT papers on average 1,623.

(b) The CAUSALCITE values of three example ToT papers from general AI, NLP, and CV.

Figure 2: Correspondence of our CAUSALCITE metric and ToT papers.

observe in Table 3 that even though some domains have significantly more citations (for instance, CV ToT papers have, on average, 4.05 times more citations than NLP), the ACI remains consistent across various fields.

## 7 FINDINGS

Having demonstrated the effectiveness of our metrics, we now explore some open-ended questions: (1) Do best papers have a high causal impact? (Section 7.1) (2) How does the CAUSALCITE value distribute across papers? (Section 7.2) (3) What is the impact of some famous papers evaluated by CAUSALCITE? (Section 7.3) (4) Can we use this metric to correct for citations? (Section 7.4).

### 7.1 DO BEST PAPERS HAVE A HIGH CAUSAL IMPACT?

Using the same set of conferences as the ToT papers, we conduct a small-scale study on 20 best papers and a control set of random non-best papers from the same conference in the same year. We find that the correlation of the CAUSALCITE metric with best papers is 0.298, which is very low compared to the 0.623 correlation with the ToT papers. This shows that the best papers do not necessarily have a high causal impact. One interpretation can be that the best paper evaluation is a forecasting task, which is much more challenging than the retrospective task of ToT paper selection.

### 7.2 WHAT IS THE CURVE SHAPE OF THE CAUSALCITE DISTRIBUTIONS?

We show the distribution of TCI in Figure 3, where we can see a power law distribution with a long tail, echoing with the common belief that the paper impact follows the power law, with high impact concentrated in a relatively small portion of papers.

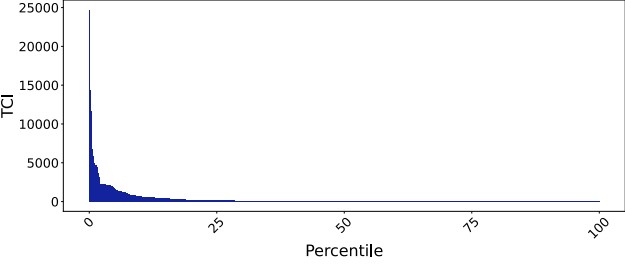

Figure 3: The distribution of TCI values by percentile, which shows a long tail indicating that high impact is concentrated in a relatively small portion of papers.

| Paper Name | TCI | Citations | ACI |
|---|---|---|---|
| Transformers | 52,507 | 68,064 | 0.771 |
| BERT | 40,675 | 59,486 | 0.683 |
| RoBERTa | 6,932 | 14,434 | 0.480 |
| MNLI | 2,475 | 3,165 | 0.782 |
| GLUE | 3,459 | 4,400 | 0.786 |
| SuperGLUE | 716 | 1,427 | 0.501 |

Table 4: Case study of some famous dataset and model papers.

### 7.3 FAMOUS PAPER CASE STUDY

In addition to the overall distribution of our CAUSALCITE values, we also look at its correspondence to some famous papers. We show in Table 4 the causal impact index values for a set of famous papers. For example, we know that the Transformer paper (Vaswani et al., 2017) is a more foundational work than its follow-up work BERT (Devlin et al., 2019), and BERT is more fundational than its later variant, RoBERTa (Liu et al., 2019). And we can see this monotonic trend in the TCI and ACI values of the three papers too. Again, this is a preliminary case study, and we welcome future work to Cover more papers.

### 7.4 DISCOVERING GOOD PAPERS LESS RECOGNIZED BY CITATIONS

Another important contribution of our metric is that it can help discover papers that are traditionally overlooked by citations. To achieve the discovery, we formulate the problem as outlier detection, where we first use a linear projection to handle the trivial alignment of citations and CAUSALCITE, and then analyze the outliers using the interquartile range (IQR) method (Smiti, 2020). See the exact calculation in Appendix C.1. We show the three subsets of papers in Table 5, where the overcited and undercited paper The two outlier categories correspond to the false positive and false negative oversight by citations, respectively. An additional note is that, when we look into some characteristics of the three categories, we find that the citation frequency in result section, i.e., the percentage of times they are cited in results section compared to all the citations, correlates with these categories. Specifically, we find that the undercited papers tend to have more of its citations concentrated in the results section, which usually indicates that this paper constitutes an important baseline for a the follow-up study, while the overcited papers tend to be cited out of the results section, which tends to imply a less significant citation.

| Paper Category | Citations in Result Section (%) | Residual |
|---|---|---|
| Overcited Papers (7.04% of the Papers) | 1.26 | -1.792 |
| Aligned Papers (91.20%) | 1.51 | 0.118 |
| Undercited Papers (1.76%) | 1.90 | 1.047 |

Table 5: We use our CAUSALCITE metric to discover outlier papers that are overlooked by citations. We find that the undercited papers (with high TCIs, but fewer citations than what they should have been deserved) are usually more cited in the results sections, which indicates a more significant role in the follow-up papers.

## 8 CONCLUSION

In this study, we propose CAUSALCITE, a novel causal formulation for paper citations. Our method combines traditional causal inference methods with the recent advancement of NLP in LLMs to provide a new causal outlook on paper impact by answering the causal question: "Had this paper never been published, what would be the impact on this paper's current follow-up studies?". With extensive experiments and analyses using expert ratings and test-of-time papers as criteria for impact, our new CAUSALCITE metric demonstrates clear improvements over the traditional citation metrics. Finally, we use this metric to investigate several open-ended questions like "Do best papers have a high causal impact?", conduct a case study of various famous papers, and suggest future usage of our metric for discovering good papers less recognized by citations for the scientific community.

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

# Appendix

## A    ADDITIONAL IMPLEMENTATION DETAILS

### A.1    NUMERICAL ESTIMATION METHOD: FINDING THE SAMPLE SIZE

For our numerical estimation method, we first calculate the ACI on a subset of carefully sampled papers and then aggregate it to TCI. To decide the size of this subset, we need to balance both the computation time (25 minutes per paper) and the estimation accuracy. To find the best balanced number, we conduct a small-scale study, first obtaining the TCI using our upper-bound budget of $n = 100$ samples and then gradually decreasing the number of samples to see if there is a stable point in the middle which also leads to a result close to that obtained with 100 samples. As in Figure 4, we can see an elbow around n=40, which is relatively close to the result when $n = 100$, but vastly saving our computers and budget so that we can run efficient experiments on more papers.

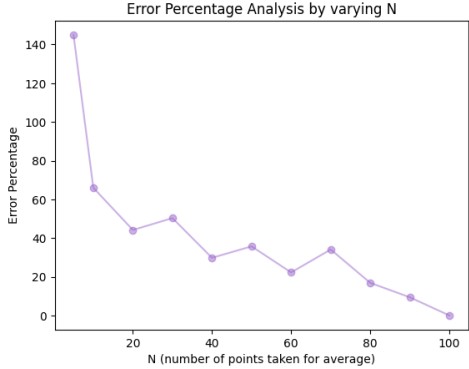

Figure 4: TCI error percentage buy varying the sample size n=10,20,...,100.

### A.2    EXPERIMENT TO SELECT THE BEST EMBEDDING METHOD

When we compare among the three LLMs pre-trained on scientific papers, SciBERT, MPNet, and SPECTER, we conduct a small scale experiment to see how much the similarities scores based on the embedding of each model align with human annotations. As for the annotation process, we first collect a set of random papers as pivot papers, and for each pivot paper, we identify 10 papers, from the most similar to the least, with monotonically decreasing similarity. We shall an example of such a collection in Table 6. Then we see how the resulting similarities scores conform to this order. We report the percentage of papers that are out of place in the ranking, and find that MPNet performs the best.

| Paper Index | Title | SciBERT | MPNet | SPECTER |
|---|---|---|---|---|
| *Pivot Paper: GPT3* | | | | |
| e1 (Most similar) | PaLM | 0.9787 | 0.7679 | 0.8689 |
| e2 | GPT2 | 0.9346 | 0.8196 | 0.9064 |
| e3 | GPT | 0.9488 | 0.779 | 0.8778 |
| e4 | BERT | 0.943 | 0.6784 | 0.8321 |
| e5 | Transformers | 0.9202 | 0.6385 | 0.8644 |
| e6 | SciBERT | 0.8396 | 0.5667 | 0.8112 |
| e7 | Latent Diffusion Models | 0.9586 | 0.4567 | 0.7755 |
| e8 | Sentiment analysis (DL) | 0.7775 | 0.2911 | 0.7298 |
| e9 | Sentiment analysis (ML) | 0.6462 | 0.2563 | 0.6403 |
| e10 (Least similar) | New High Energy Accelerator | 0.8033 | 0.0359 | 0.5617 |

Table 6: Comparison for the three embedding methods.

### A.3    MORE IMPLEMENTATION DETAILS

For the time cost of running the causal impact indices, each $PCI(a, b)$ takes around 1,500 seconds, or 25 minutes. Multiplying this by 40 samples per paper $a$, we spend 16.67 hours to calculate each

ACI or TCI for the paper's overall impact. For a fine-grained division into the time cost, the majority of the time is spend on the BM25 indexing (800s) and the sentence embedding cosine similarities calculation (400s). The rest of the time-consuming steps are the BFS search (150-200s every time) to identify descendants and non-descendants of a paper.

For the space complexity, we loaded the 2.4B edges of the citation graph into a parquet gzip format for faster loading, and use Dask's lazy load operation to load it part by part to RAM for better parallelization. The program can fit into different sizes of RAMs by modifying the number of partitions and reducing the number of workers in Dask, at the cost of an increased computation time. On the hard disk, citation graph takes up 19G space, and paper data takes 11G.

## B DATASET OVERVIEW

When using the Semantic Scholar dataset (Kinney et al., 2023; Lo et al., 2020), we deal with the paper data and citation network separately. For the 206M paper data, we obtain them using the "Papers" endpoint to get the Paper Id, Title, Abstract, Year, Citation Count, Influential Citation Count by (Valenzuela et al., 2015), Reference Count for each paper. The papers come from a variety of fields such as law, computer science, and linguistics. For the citation network with 2.4B edges, we use the Semantic Scholar Citations API to get each edge of the citation graph in the form of the (fromPaper, toPaper, isInfluentialCitations) triple.

From Figure 5, we can see that, in general, the number of publications explode in recent years. The left plot shows the number of papers publish the per year, which reaches on average 7.5M per year since 2010. The right plot shows the number of references a paper cites, which also increases from less than five before 1970s, to around 25 in recent years.

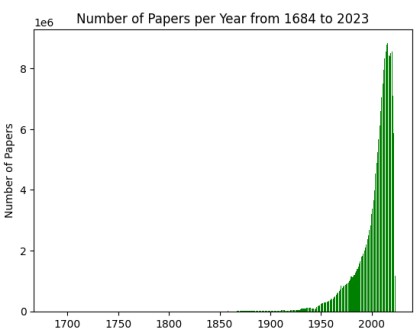 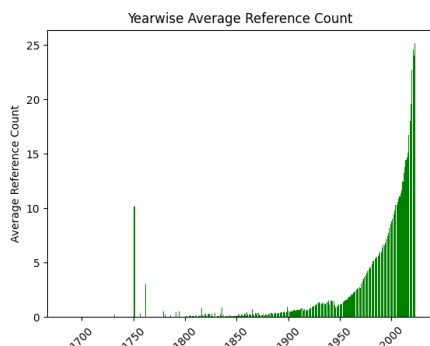

Figure 5: The left figure illustrates the number of papers published per year. The dataset spans from 1684 to 2023, and in recent years since 2010, there are more than 7 million papers each year. Right figure shows the Yearwise average of the number of references per paper. The plot shows an YoY increasing trend which demonstrates the need of our paper. Since the average number of papers referenced in papers per year is increasing, researchers need a metric to sort the most influential references so that they can get an idea by reading those.

## C ADDITIONAL ANALYSES

### C.1 CITATION OUTLIER ANALYSIS

For the outlier detection, we first to visualize the scatter plot between our CAUSALCITE and citations in Figure 6. Then, we fit a log-linear regression to learn the line $\log(\text{TCI}) = 1.026 * \log(\text{Cit}) - 0.541$, with a root mean squared error (RMSE) of 0.6807. After fitting the function, we use the interquartile range (IQR) method (Smiti, 2020), which identify as outliers any samples that are either lower than the first quartile by over 1.5 * IQR, or higher than the third quartile by more than 1.5 * IQR, where IQR is the difference between the first and third quartile.

We denote as overcited papers the ones that are identified as outliers by the IQR method due to too many citations than what it should have deserved given the CAUSALCITE value. Symmetrically, we denote as undercited papers the ones that are identified as outliers by the IQR method due to too

few citations than what it should have deserved given the CAUSALCITE value. And we denote the non-outlier papers as the aligned ones.

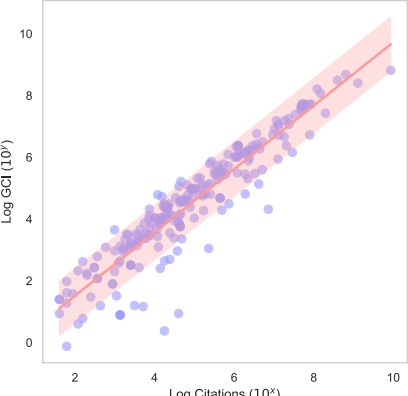

Figure 6: The scatter plot between our CAUSALCITE and citations, with the fitted function as $\log(\text{TCI}) = 1.026 * \log(\text{Cit}) - 0.541$, and a non-outlier band width of 0.8809.

