# OpenReview forum: "Causal Impact Index: A Causal Formulation of Citations"
_ICLR.cc/2024/Conference — ICLR 2024 Conference Withdrawn Submission_

### Official Review · Reviewer_tkEL · 2023-10-23

**Soundness:** 2 fair
**Presentation:** 2 fair
**Contribution:** 2 fair
**Rating:** 3
**Confidence:** 4

**Summary:**

The authors propose a causal impact index which measures how influential a paper is via counterfactual estimation.

**Strengths:**

Thanks to the authors for the hard work on this paper.

**Weaknesses:**

- I don't really understand the method. A lot of specific questions are in the next section, but I am missing the big picture. Paper $b$ cites paper $a$. When you find nearest neighbors of paper $b$, do you make sure that none of them *also* cite paper $a$? Do you make sure they are from the same year? Also how can you tell that they are truly similar enough for the estimation to be legitimate? Is there some theoretical requirement that these nearest neighbors to $b$ are close enough?

- Unfortunately "average 20-30 minutes for each PCI(a, b)" is too slow to be adopted at scale as a substitute for citations or influential citations or other metrics

- Also, it makes sense to compare to other alternatives besides the ones you have already, e.g. https://arxiv.org/abs/2102.03234
and https://arxiv.org/abs/2105.08089

Minor:
- "we consider LLM variants pretrained on large-scale scientific text, such as SciBERT (Beltagy et al., 2019b), SPECTER (Cohan et al., 2020), and MPNet (Song et al., 2020)." These are too small to be called LLMs. Safe to call them LMs.
- Figure 3 should have a log y-axis

**Questions:**

- "we first exclude expressions about the quality or performance of the paper, such as the notion “state-of-the-art” and the exact performance numbers in arabic numbers" - how? How good is this procedure? Please evaluate this step, even with a small scale manual evaluation.
- "we bin the large set of follow-up papers into n equally-sized intervals" - bins of what? Citation counts?
- Algorithm 1: "non-follow-up papers C" - what are these? How do you get them? The `getPCI` method uses them. What is this method?
- Algorithm 1: as far as I can tell this is just "iterative averaging over a random subset sampled from bins". Right? If so, it doesn't need its own algorithm box. And it doesn't seem "novel" enough to warrant that label.
- "For the set of matched papers, we consider papers with cosine similarity scores higher than 0.93" - How did you arrive at this value? Was it tuned to increase the score in Table 1? I am worried about overfitting.
- What if none of the nearest neighbors are closer than 0.93?
- What about citations of citations? If paper $a$ was cited first, and then paper $b$ cited it, and then paper $c$ can either cite $a$ or $b$ but often not both. Does this need to be controlled for?

---

### Official Review · Reviewer_Hvdi · 2023-11-01

**Soundness:** 2 fair
**Presentation:** 3 good
**Contribution:** 2 fair
**Rating:** 3
**Confidence:** 4

**Summary:**

In this work, authors devised an alternative index to citation count, namely CAUSALCITE, through a causal lens. The goal is to handle the limitations and weakness of the current index and ultimately measure the true impact of academic paper. In action, authors used LLM to extract the representation of papers. Then, for each paper, they generated a counterfactual paper via weighted KNN, and then calculate the causal effect. Intersting case studies have been conducted to evaluate the efficacy of the proposed approach.

**Strengths:**

1. The research problem is important and challenging: how to measure the true impact of academic paper?
2. It is innovative to formulate this research problem as a causal inference problem. It is also interesting to leverage causal graph to analyze the roles of each factor, identify the potential bias and devise adjustment-empowered estimators.
3. The paper is easy to follow, with impressive coherence and narrative.

**Weaknesses:**

The concerned limitation of citation count is not very clear. The terminology used, such as "failing to accurately reflect a paper’s true impact," offers intuitive but vague criticisms of the existing citation-count system. Moreover, the authors do not adequately address how CAUSALCITE overcomes these issues. Explicitly, questions like "Is it feasible to formulate the limitations (especially the bias) of citation count in a causal graph?" and "How does CAUSALCITE solve the claimed limitations?" remain unanswered. Providing such information would help clarify the scope and the innovative aspect of the CAUSALCITE model.

Lack of limitation. Dicussion on limitation is very important especially for a new metric since each metric as its pros and cons. Although the authors did not involve this point in the main text, after rethinking I suggest some easy but common cases where CAUSALCITE could fail. Authors could formulate them as assumptions or reflect them in causal graphs, to clarify the scope of this work. Authors are highly encouraged to debate for their completeness, but an index could not be excellent in every aspect. Identifying and formulating the limitations could largely improve the quality of the paper.

- Data sparsity in high-dimensional spaces. Even though cosine similarity is effective in such spaces, if there are not enough 'near neighbors' for a given paper, the method could fail to find adequate matches. This would result in counterfactual samples that are not truly representative, affecting the validity of the causal inference. Imagine trying to synthesize a counterfactual for a groundbreaking paper on a niche subject; the lack of similar papers would make this task challenging.

- Temporal dynamics. Academic papers are not static entities; they gain citations over time, undergo revisions, and may be commented upon. If the method does not account for these temporal changes, the synthesized counterfactual may be not accurate. For instance, a paper published ten years ago that has been highly cited will have different characteristics than a similar paper published recently, and simply averaging the two could produce misleading results.

- Dimension reduction bias when encoding papers using text embeddings. Although LLMs can capture some textual features, the embeddings are still a reduced representation of the original text. Thus, they might not capture all nuances or specialized details, and **numerical results** present in the paper, which could be key factors for accessing the paper quality.
For example, two papers focusing on a topic, with the same experimental pipeline, could derive different and even contractory results (which is the usual case in some subjects...). Can such difference be reflected by LLM embeddings?


Lack of theoretical backup. The core technical point seems a weighted KNN. Since there are many causal inference methods, such as re-weighting-based, matching-based and representation-based methods, the motivation to select weighted KNN to adjust data should be formulated.  As a causality-inspired paper, theoretical formulation and backup is a critical aspect. Is there any theoretical formulation of the advantages of CAUSALCITE over citation count? In which conditions? Moreover, identifiability is a crucial aspect in causality empowered methods. Is CAUSALCITE identifiable? Is there any conditions to support the identifiability of CAUSALCITE?

**Questions:**

Please see the questions in the weaknesses section.

---

**My rating only reflects the current status of this manuscript. Since the research question is honestly impressive, at least for me, I am willing to rethink my rating if the authors could mitigate my main concerns (weaknesses 2,3).** The authors are highly encouraged to refute any inaccuracies in my comments directly.

---

### Official Review · Reviewer_kS1d · 2023-11-06

**Soundness:** 2 fair
**Presentation:** 3 good
**Contribution:** 2 fair
**Rating:** 3
**Confidence:** 3

**Summary:**

In this paper, the author proposed a framework for understanding the citation of a paper by analyzing the citation graph from a causal perspective.

**Strengths:**

1. The paper is well-written and easy to follow.

2. The problem setting is solid and definitely interesting, that is to understand the counterfactual effect of a paper not being published.

**Weaknesses:**

1. The evaluation counts on the construction of the synthetic dataset, and the reliability of the counterfactual data is not measured or analyzed.

2. The assumption that the confounders are " a paper's publication year, research area, research question, and storyline" is limited. These can be part of the confounder and the storyline is a concept that is hard to measure.

 3. To measure the impact of people's work, there might be a need to provide a more rigorous justification, especially when using the LLM for synthetic data generation.

**Questions:**

See weakness.

---

> ### Public Comment · ~ZJin1 · 2023-11-23
>
> Thank you for your feedback that the paper is well written, easy to follow, and has a solid an interesting setting.
>
> > 1. The evaluation counts on the construction of the synthetic dataset, and the reliability of the counterfactual data is not measured or analyzed.
> > 3) To measure the impact of people's work, there might be a need to provide a more rigorous justification, especially when using the LLM for synthetic data generation.
>
> Regarding the two points raised about the synthetic dataset and the use of counterfactual data, there seems to be a misunderstanding. Our paper does not involve the generation of synthetic data using LLMs. Instead, we employ text embeddings to match real academic papers to other real papers. Here's a brief outline of our methodology and evaluation:
>
> - **Methodology**: We have innovated the method of matching in causal inference, adapting it for real-to-real paper matching. For a given paper, we identify a set of similar papers based on content and then obtain the weighted average of their citation scores. This is all based on actual, real-world papers.
>
> - **Evaluation**: For the result evaluation, we also conduct the experiments on real papers, using a real dataset of 200 million papers from semantic scholar, and also evaluating it on 1K annotations of real papers' impact on each other. We further diversified the evaluation from three different aspects, all focusing on real papers.
>
> > The assumption that the confounders are " a paper's publication year, research area, research question, and storyline" is limited. These can be part of the confounder and the storyline is a concept that is hard to measure.
>
> Thank you for the suggestion. To address the problems you proposed, our approach goes beyond traditional low-dimensional data matching methods. Instead of relying solely on small vectors to encode these dimensions (which might inadequately capture complex features like storylines), we utilize high-dimensional text embeddings. These embeddings effectively encapsulate the rich semantics of a paper, thereby overcoming the limitations of an incomplete covariate list. This methodological choice is a significant strength of our work, allowing for a more nuanced and comprehensive understanding of academic papers.
>
>
>
> ---
>
> **In Conclusion**
>
> We hope these clarifications address your concerns. We would greatly appreciate a re-evaluation of our contributions, which we believe are significant:
>
> 1. **Method Innovation**: Our use of text embeddings for the traditional causal inference method of matching is a novel contribution to the field.
>
> 2. **Practical Application**: We propose an impactful use case in making paper citations more causally attributable, thus offering a fairer system of academic credit.
>
> 3. **Robust Evaluation**: Our methodology's effectiveness is validated using real papers, and we compare our results with human annotations of paper impact. Our approach shows a clear improvement over existing citation metrics, making it a valuable contribution to the academic community.
>
> We look forward to your reconsideration and remain open to further discussion.